# Could a Conversational AI Identify Offensive Language? †

**Daniela America da Silva** *,‡ , **Henrique Duarte Borges Louro** ‡ , **Gildarcio Sousa Goncalves** ‡ , **Johnny Cardoso Marques** ‡ , **Luiz Alberto Vieira Dias** ‡ , **Adilson Marques da Cunha** ‡ and **Paulo Marcelo Tasinaffo** ‡

Electronic and Computer Engineering Program, Informatics, Brazilian Aeronautics Institute of Technology, ITA, Sao Jose dos Campos 12228-900, Brazil; henrique.dblouro@gmail.com (H.D.B.L.); gildarcio@ita.br (G.S.G.); johnny@ita.br (J.C.M.); l_vdias@yahoo.com.br (L.A.V.D.); cunha@ita.br (A.M.d.C.); tasinaffo@ita.br (P.M.T.)
* Correspondence: damerica@ita.br
† This paper is an extended version of our paper published in Information Technology: New Generations.
‡ These authors contributed equally to this work.

**Abstract:** In recent years, we have seen a wide use of Artificial Intelligence (AI) applications in the Internet and everywhere. Natural Language Processing and Machine Learning are important subfields of AI that have made Chatbots and Conversational AI applications possible. Those algorithms are built based on historical data in order to create language models, however historical data could be intrinsically discriminatory. This article investigates whether a Conversational AI could identify offensive language and it will show how large language models often produce quite a bit of unethical behavior because of bias in the historical data. Our low-level proof-of-concept will present the challenges to detect offensive language in social media and it will discuss some steps to propitiate strong results in the detection of offensive language and unethical behavior using a Conversational AI.

**Keywords:** offensive; dictionary; natural language; AI ethics; fairness





## 1. Introduction

Currently, algorithms with Artificial Intelligence (AI) origins are currently used for many tasks in a variety of domains. Although the computer learning ability is still inferior to human learning ability [1], there is a demand for Machine Learning (ML) algorithms as one of the most important branches of AI. Nowadays, ML algorithms have been used in voice recognition systems, spam filters, online fraud detection systems, product recommendation systems, in education, and in many different areas.

A crucial area in AI is the Natural Language Processing (NLP) because it models how people share information and how to build systems to keep conversations with humans. Jurafsky in Chapter 24 of the book Speech and Language Processing [2] presents that language is the hallmark of human conversation and our ability to be aware of feelings and sensations. The conversation or dialogue is the most privileged part of the language and it is the first type of speech we learn as children and it is applied when, for example, we are asking or buying something, attending meetings, talking to our families, complaining about the weather, and various other activities.

However, human conversations has specific characteristics, for example, turns, speech acts, grounding, dialogue structure, initiative, and implicature, and those are some of the reasons why it is difficult to build dialogue systems that can carry on natural conversations with humans [2]. In addition, build dialogue systems that can identify offensive language and/or unethical behaviors on natural conversations is even more complex because offensive language and/or unethical behavior is an action that falls outside what is considered morally right or proper for a person, a profession or an industry, however it can take multiple forms, cultural characteristics and targets that are difficult to model.

A misunderstanding about Chatbots and Conversational AI may make people think that Chatbots are prepared to human-to-human interactions, however they are designed to

extended conversations and to mimic characteristics of human-to-human and could be not prepared to notice offensive language and/or unethical behaviors.

In addition, the AI algorithms are built based on historical data and specifically in Natural Language Processing. However, historical data could be intrisically discriminatory and a large amount of data is necessary, in order to create language models.

A Conversational AI is composed by end-to-end spoken language understanding (SLU) models to predict semantics directly from speech [3]. The conventional approach to SLU uses two distinct components to sequentially process a spoken utterance: an automatic speech recognition (ASR) model that transcribes the speech to a text transcript, followed by a natural language understanding (NLU) model that predicts the domain, intent, and entities given the transcript. Recent applications of deep learning approaches to both ASR and NLU have improved the accuracy and efficiency of SLU systems and driven the commercial success of voice assistants such as Amazon Alexa and Google Assistant. As SLUs also use a speech transcribed to a text, it could be also used to address offensive language considering also bias and discrimination.

In this article, we investigate the advances in algorithms used on natural conversations and if they could identify offensive language and/or unethical behaviors. This work advances in the analysis of current problems of offensive language identification. Moreover, it is an extension of a previous publication called "A Hybrid Dictionary Model for Ethical Analysis" [4], checking its applicability in a low proof of concept to identify offensive language in Portuguese.

First, in the Background section we demonstrate the need for offensive language detection in Conversational AI. Afterwards, the section Methods describes the creation of a dictionary for offensive language detection explaining the evolution from previous work about a hybrid dictionary model for ethical analysis [4].

In the Results section, we describe the application of the Dictionary to a text corpus to identify offensive language, applying the dictionary to Twitter posts. Moreover, this will work as a low-level proof of concept. Prior to the finalization, the Discussion section will focus on the results of this paper in light of the research landscape as described in the Background section, and it will discuss some steps to propitiate strong results in the detection of offensive language and unethical behavior using a Conversational AI. Finally, the Conclusion section presents the main achievements and also make recommendations about the work conducted.

## 2. Background

The general objective of this work is to raise questions about whether and how Chatbots and Conversational AI can handle unethical user behavior. It will also present a low-level proof-of-concept for detecting offensive language in social media. This study followed a systematic mapping method [5,6], in order to present an overview of a research area to report the amount and type of literature and results that are published in it. The systematic mapping process comprises three steps: (1) the identification of relevant literature, (2) the composition of a classification scheme, and (3) the [5] literature mapping. In this way, the following mapping questions were defined:

1. Chatbots and Conversational AI can identify offensive language?
2. Could ML be designed for reliability and justice?
3. Do the ethical issues depend on the geographical area?

Because the field of Chatbots and Conversational AI is relatively new, the most important, transformative, and relevant articles are very recent. In this study, articles have been recommended by specialists, searched in the Scopus and Web of Science databases, in important conferences such as ACM FAT, NeurIPS, ICML, and AAAI, and also in certification courses such as in UMontrealX and IVADO, Bias, and Discrimination in AI (MOOC from edX [7,8]). The eligibility criteria was to identify documents discussing AI Ethics, AI Bias and Discrimination, Machine Learning Fairness, Language Models, and solutions to detect offensive language and/or unethical behavior. Afterwards, the

documents were organized into specific areas for analysis. In total, it was selected 2 books, 24 articles, and one guide. A summary on documents selected is available in Table 1 and the details about the systematic map flow are presented in Figure 1.

**Table 1.** Articles selected in the systematic mapping.

| Areas | Documents |
|---|---|
| AI Ethics: | IEEE [9], Jobin et al. [10], Hutson [11], Awad et al. [12] |
| AI Bias & Discrimination: | Suresh and Guttag [13], Tufekci [14], Olteanu et al. [15], Caliskan et al. [16] |
| Fair ML: | Hutchinson and Mitchell [17], Verma and Rubin [18], Mehrabi et al. [19], Liu et al. [20], Zhang and Ntoutsi [21], Zhang et al. [22], Bechavod et al. [23] |
| Language Models: | Jurafsky and Martin [2], Bommasani et al. [24], Abid et al. [25], Gebru et al. [26], Bender and Friedman [27] |
| Offensive Language: | Tauszcik and Pennebaker [28], Mondal et al. [29], Chiu and Alexander [30], Gordon et al. [31], Sap et al. [32], Davidson et al. [33], Davidson et al. [34] |

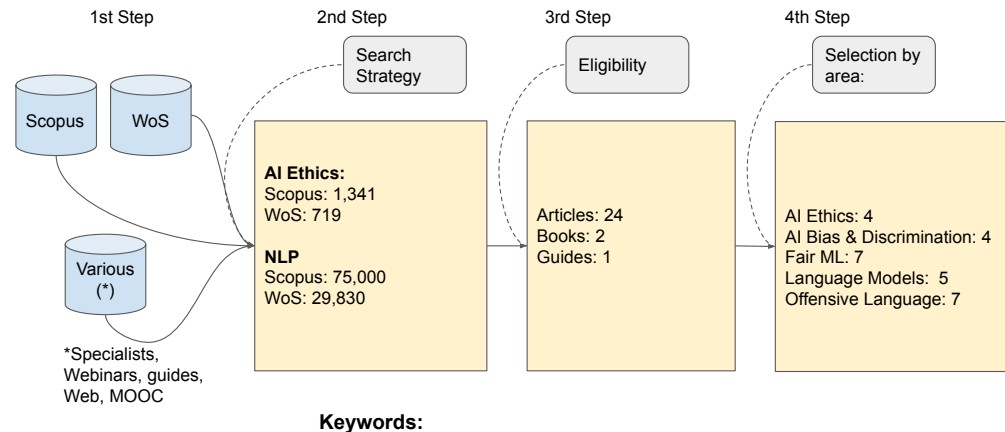

**Figure 1.** Systematic mapping flow.

### 2.1. AI Ethics

The AI performance is directly linked to the quantity, quality, and representativeness of data, and developers need training data that resemble all the diversity and complexity of the real world operating environment [35]. This new generation of data-intensive technologies will require, in addition to current digital skills, the awareness to understand that data are never just raw. New discussions in areas such as data protection, transparency, explainability, justice, responsibility, partiality, quality, and ethics between different organizations in society will be increasingly relevant [36].

Given the impact of the use of algorithms, IEEE developed a document called IEEE Ethically Aligned Design (EAD) [9], to guide the development of autonomous and intelligent systems, with the participation of 800 engineers. IEEE currently has more than 420,000 members.

The IEEE EAD guide addresses classical ethical methodologies in algorithm design considerations for Autonomous and Intelligent Systems (A/IS) where ML may or may not reflect ethical results that mimic human decision making. A comprehensive summary of contributions from classical ethics for the design of A/IS is presented as follows.

1. Virtue Ethics —Aristotle argues that a moral agent reaches "flowering", seen as an action and not a state, through the constant balance of factors as the social environment, material provisions, friends, family, and yourself. In the context of A/IS, there

are two immediate values of virtue ethics: a model for iterative learning and growth; a framework for A/IS developers counteracted tendencies towards excess.

2. Deontological ethics—Developed by the German philosopher Immanuel Kant, it is based on moral and legal responsibility. In other words, a rule must be inherently desirable, feasible, valuable, and others must be able to understand and follow it. Rules based on personal choices cannot be universalized. In the context of A/IS, the question is whether developers are acting with the best interests of humanity and human dignity in mind.

3. Utilitarian (Consequentialist) ethics—Refers to the consequences of your decisions and actions, that is, the right course of action is the one that maximizes utility (utilitarianism) or pleasure (hedonism) for the greatest number of people, but considering superficial and short-term usefulness or pleasure. In the context of ethical AI, it is the responsibility of A/IS developers to consider the long-term effects including social justice.

4. Ethics of Care—This philosophy emphasizes the importance of relationships, and taking care of another human being is one of our basic attributes. That is, the relationship with another person must exist or have the potential to exist and the relationship should be one of growth and care. With regards to A/IS, if A/IS is expected to be beneficial to humans then the human will need to take care of A/IS. Moreover, if this possibility exists, then principles applicable to A/IS will be needed.

The IEEE EAD guide also explores contributions from ethical systems based on religion and culture and explains that many non-Western traditions see "relationship" as a fundamental concept for discussions of ethics. The guide suggests a special focus on similarities in the cross-cultural understanding of the concept of "relationship" as it could complement the discussion of ethical issues for A/IS.

Also in this direction on the impact of cultural aspects for autonomous systems, an MIT Lab experiment called The Moral Machine Experiment verified the preferences of individuals in the use of autonomous cars, and this experiment argues that there are global preferences and individual and cultural clusters in the use of systems, among them: Western cluster (Protestant, Catholic, and Orthodox Christian cultural groups), Eastern Cluster (Confucian cultural groups and Islamic countries), and Southern Cluster (Latin America is composed by sub-clusters, French, Portuguese, and Spanish). Additionally, geographic proximity can make clusters converge on ethical issues, but there may be internal differences in each cluster [12].

In the field of NLP, some research investigations are verifying that current algorithms are still not close to what a human would respond because a human does not respond based on historical statistical data, the human responds based on its iteration with the environment [11].

With regards to metrics and based on ethical principles, the IEEE guideline proposes three very important metrics, when developing A/IS systems, as listed below.

1. Transparency—the basis of the decision-making process of an autonomous and intelligent system (A/IS) must be always detectable.

2. Responsibility—an A/IS must be created and operated according to an unambiguous logic for all decisions taken.

3. Awareness of misuse—the A/IS creators must protect themselves against all possible undue, inequitable, and risks of the A/IS in operation.

There are significant technical and early-stage efforts to respond to current practical problems and provide an ethical AI, and in recent years companies have invested significant resources into ethical AI, that is, they have established metrics to detect and remove problems that may be caused by AI [10].

For example, the Principles Approach from the Berkman Klein Center Institute at Harvard University [37] brings all the guidelines created since 2016, segregating by contributions, civil society, government, intergovernmental, private sector, and multi-stakeholder.

The good practice of IEEE EAD [9] is classified as multi-actor. In Figure 2, we present an overview on the principles developed adapted from the Berkman Klein Center Institute at Harvard University.

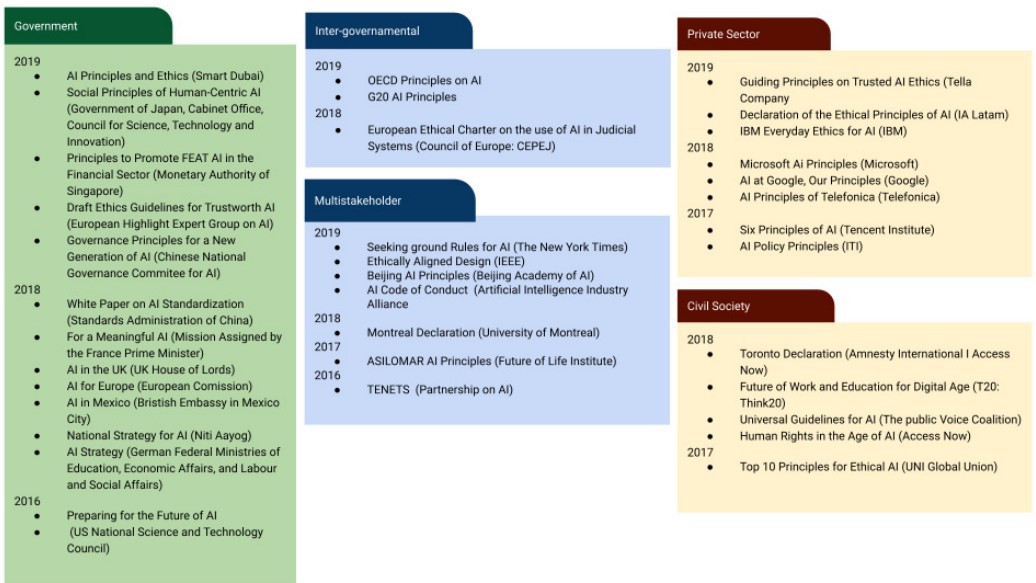

**Figure 2.** Summary on AI Principles according to Berkman Klein Center [37].

In particular, discussions need to be stimulated in Latin America to respond to specific language needs and social characteristics. The region has a large pool of people with soft-skills, problem solving, young people, collaborative, and creative leadership to drive the ethical use of AI.

### 2.2. AI Bias and Discrimination

Algorithms are used differently than human decision makers, however frequently people assume that algorithms are objective or error-free. Furthermore, algorithms are often cheap, used at scale, and more likely to be implemented without process in place. As humans are biased, many people may think that bias in algorithms is not a problem. However, machine learning algorithms can amplify bias and create feedback loops not always consistent with the reality. Therefore, there is a lot of responsibility involved in using this powerful technology.

When bias is incorporated in our decisions it becomes discrimination. There are different types of discrimination: direct (e.g., when someone refuses to rent an apartment because of person origin), indirect (e.g., when applying a neutral rule with harmful effects), and systemic (e.g., offer lower wages to a woman than to men) [7].

Olteanu, A. et al. [15] addresses general challenges for social data use, which include the following areas and definitions: (i) Data bias, a systematic distortion in the sampled data that compromises its representativeness. (ii) Population biases, systematic distortions in demographics or other user characteristics between a population of users represented in a dataset or on a platform and some target population. (iii) Behavioral biases, systematic distortions in user behavior across platforms or contexts, or across users represented in different datasets. (iv) Content production biases, behavioral biases that are expressed as lexical, syntactic, semantic, and structural differences in the content generated by users. (v) Linking biases, behavioral biases that are expressed as differences in the attributes of networks obtained from user connections, interactions or activity. (vi) Temporal biases, systematic distortions across user populations or behaviors over time. (vii) Redundancy, single data items that appear in the data in multiple copies, which can be identical (duplicates), or almost identical (near duplicates) and can distort quantification of phenomena

in the data. As presented in those general challenges, there are a lot of complexities in addressing bias and there is no one-size-fits-all solution.

Continuing the understanding of how bias can impact algorithms, Suresh H. et al. [13] identifies seven sources of harm in ML building and implementation: (i) Historical bias, happens even when data is perfectly measured and sampled, because it leads to a model that produces harmful outcomes, for example, reinforcing a stereotype. (ii) Representation bias, occurs when development sample under-represent some part of the population. (iii) Measurement bias, it happens when a feature and labels are chosen to approximate some idea that is not encoded or observable, for example, credit score can be problematic if they poorly reflect groups. (iv) Aggregation bias, arises when it is assumed the map from inputs to labels is consistent, however there are subgroups that should be considered differently. (v) Learning bias, occurs when modeling amplifies disparities across different examples in the data. (vi) Evaluation bias, happens when data used for a particular task does not represent the use population. (vii) Deployment bias, when there is a mismatch between the problem a model is intended to solve and the way it is used.

To address additional challenges in social data use, Tufekci, Z. et al. [14] verify the validity and representativeness of social media big data. The study explains that there are a few platforms frequently used to generate datasets without adequate consideration of biases, for example, big data research focuses disproportionately in Twitter. Furthermore, analysis is conducted using hashtags that are embedded in particular cultural and sociopolitical aspects, and a hashtag could be a declaration of a particular sympathy, and therefore although useful, the cultural aspects surrounding a specific hashtag needs to be addressed. In social media it is also possible to identify who clicked, but usually it is not possible to know who saw or who could have seen a post, and the characteristics of this sub-population is rarely know. Finally, information in human affairs flows through many channels, and the Internet can not be confined to a single platform.

To demonstrate how bias can impact text, Caliskan A. [16] show how human-like semantic biases that result from application of standard ML to language humans are exposed every day. The study indicates that language itself contains our historic biases, that could be neutral (when talking about insects or flowers), problematic (when related to race or gender) or veridical (when reflecting distributions such as first names or genders and careers). The study shows that AI can and does inherit the same biases from humans. However, the impact of biases in AI is much bigger than in humans, because learning in AI can be shut off completely after a system is put in place and it could perpetuate biases in society for a long time. For example, let us say a sentiment analysis is applied for a movie review, Caliskan A.'s results show that European-American names have more positive valence than African-American names when using word embedding. Therefore, the tool will display a racial bias based on actor and character names.

It is important to clarify that bias is subjective and relative to task, for example, in health care it is not discriminative if the diagnosis is gender oriented, however in hiring it could be discriminative if it is gender-biased. Therefore, it is important to be careful and know what kind of measurement to use in a bias scenario [8].

### 2.3. Fair ML

The concept of justice in decision-making in autonomous systems involves a critical analysis of automatic learning systems for their potential to harm historically under-represented or disadvantaged groups of a population [38]. Consequently, a variety of fairness criteria have been proposed as constraints on standard ML goals. While these criteria are clearly intended to protect the disadvantaged groups of a population, they often appeal to intuition, which is a good argument for their use.

Fairness in algorithms is important to support human values, identify strengths and weakness of the system, and also track algorithms improvements over time [8]. It is hard to come up with a single definition of fairness, because there is no single agreed upon measure for discrimination and/or fairness [8].

Around 1960, there was an increased interest in testing fairness through quantitative definitions due to US anti-discrimination legislation in the areas of education and employment. Currently, the interest in fairness is due to the public interest in the use of machine learning in criminal sentences and predictive policy. Each of these eras have similar or identical ideas. Hutchinson et al. [17] recommends a broader debate on fairness and its technical and cultural causes and that the values encoded in technical definitions be made explicit.

In the study from Verma S. et al. [18] it is proposed five categories of measures for fairness: (i) statistical based on predicted outcome (group fairness or statistical parity, conditional statistical parity); (ii) statistical based on predicted and actual outcomes (predictive parity, false positive error rate balance, false negative error rate parity, equalized odds, conditional use accuracy equality, overall accuracy equality, treatment equality); (iii) statistical based on predicted probabilities and actual outcome (test-fairness or calibration, well calibration, balance for positive class, balance for negative class); (iv) similarity-based (casual discrimination, fairness through unawareness, fairness through awareness); and (v) casual reasoning (counter factual fairness, no unresolved discrimination, no proxy discrimination, fair inference).

Farnadi G. [8] proposes to group it in three categories based on legal concepts: (i) direct vs indirect discrimination; (ii) individual vs. group fairness; and (iii) explainable vs unexplainable discrimination. Direct discrimination happens when a person is treated less favorably because of one attribute, for example, postal code, and indirect discrimination happens when there is a practice, police, or rule that applies to everyone but it has worse effect in some people than others [8]. Individual fairness is measuring the discrimination of each individual. And group fairness is measuring the impact of discrimination towards a group of people. Therefore, it should have a way of grouping people [8]. And explainable discrimination relates to differences in treatment and outcomes among different groups, it can be justified and explained via some attributes in some cases. On the other hand unexplainable discrimination happens when the discrimination towards a group cannot be explained [8,19].

Following it will be presented examples of ML fairness implementation using algorithms.

Liu et al. [20] present a one-step decision-making feedback model, exposing how these decisions impact the underlying population over time. In this study, it was chosen how to assess whether a bank loan was successful. A successful loan is considered to bring profit to the bank and also increase the borrower's credit score. To evaluate the loan process, it is proposed the analysis of three frequent criteria: (i) an unrestricted bank would maximize profit, choosing limits that meet a break-even point, above which it is profitable to grant loans; (ii) another frequent criterion is demographic parity, which requires the bank to lend to both groups at an equal rate, and therefore the bank would continue to maximize profit as far as possible; and (iii) there is also equality of opportunity, which equals the true positive rates between two groups, thus requiring the bank to lend to both groups at an equal rate between individuals.

Based on the criteria mentioned, Liu et al. [20] argue that careful temporal modeling is necessary for accurately assess the impact of different justice criteria on the population. In addition, it is necessary to understand the measurement error to assess the advantages of fairness criteria over unrestricted selection. Moreover, the work also seeks to demonstrate that intuition can be an unsatisfactory way to judge the long-term impact of justice restrictions.

In essence, the work proposes to understand the causal mechanism of two variables that translate decisions into results. This can be seen as a relaxation of requirements compared to existing studies that require knowledge of sensitive attributes (such as gender, race or proxies). The paper also argues that, without a careful model of delayed outcomes, it is not possible to predict the fairness impact that a criterion would have applied as a constraint on a ranking system. However, if such accurate result model is available, it is

possible to more directly present the optimization to obtain positive results, rather than the existing justice criteria.

Differently from previous method that proposes a model to analyze delayed outcomes of algorithms and its impact on fairness, Bechavod et al. [23] propose an auditor to observe the algorithm's decisions. To ensure fairness and non-discrimination in ML based on algorithms, Bechavod et al. [23] propose that individual justice requires that similar individuals be treated similarly. The study presents an auditor who checks for violations of justice without using a parametric form, unlike most works that use specific attributes such as race and gender. The work explains that it may be difficult for humans to enunciate a precise metric of similarity between individuals. Furthermore, a similarity metric may also be inconsistent with other metrics. That said, the work proposes that an auditor observe the decisions of a learner about a group of individuals and seek to identify a violation of justice, that is, a pair of individuals that should be treated similarly by the learner.

In this work, we prove that it is possible, even without an individual fairness restriction, to learn about individual fairness through an auditor, a simple and elegant solution that bypasses the obstacles imposed by classical solutions that use similarity metrics.

On the other hand, Zhang and Ntoutsi [21] proposes to address fairness in the data stream classification. Zhang and Ntoutsi [21] explain that automated data-driven decision-making systems depend on sophisticated algorithms and data availability. Moreover, there is constant concern about how to achieve fairness and accountability in these models as historical data is often intrinsically discriminatory. That is, the proportion of members sharing one or more sensitive attributes may be greater than the proportion in the population as a whole when receiving a positive rating, which causes a lack of justice in the decision support system.

The authors explain that the basic purpose of fairness-aware classifiers is to make fair and accurate decisions. Therefore, we train a decision-making model based on historical data even with bias, so that it provides accurate predictions for future decision-making, but without discriminating people into population subgroups. The authors then propose to address the discrimination in the data stream classification.

The authors explain that the most common approaches to address the bias and discrimination problem in machine learning systems, due to the bias inherited in the data and the complex interaction between the data and the algorithms, usually follow three approaches: (i) preprocessing, modifications can be made to the distribution of data to ensure fair representation of different communities in the training data; in this way the classifier is trained on discrimination-free data, however this approach cannot eliminate the discrimination that may come from the algorithm itself; (ii) in-processing, the modification is performed in the algorithms to consider fairness rather than just the performance of prediction; an example is to use a Naive Bayes classifier with three approaches; in the first approach, the distribution decision is altered until non-discrimination is achieved; in the second approach, a separate model is built for each sensitive group; and in the third approach, a latent variable is created to discover the discrimination-free class labels; (iii) post-processing, it modifies the result of the models by correcting decisions that could harm the fair representation of different subgroups in a final decision process.

Unlike these existing methods, the authors investigate an approach to dealing with changes in joint data over time, integrating an algorithm-level solution for a fair classification and the online approach for keeping an accurate and up-to-date classifier to infinity data streams with non-stationary distribution and bias discrimination. The model is based on Hoeffding tree to make confident decisions about the selection of splitting attributes over infinitive streams and to accommodate new instances from the stream incrementally. The model not only incorporates new examples from the stream, but also alleviates their bias towards the favorite group by applying a new split criterion, the fair information gain, which considers both information and fairness gain in a division.

Different from previous methods, Zhang et al. [22] verify if fairness could be verified online. Zhang et al. [22] clarify that many works are focused on offline data processing,

but in real-life a lot of application data is online and needs to be processed in real time. Additionally, fairness and accuracy need to be considered; however, many algorithms have hyperparameters whose iteration is not trivial to achieve fairness. This study analyzes how to adjust the components of an online classifier, as well as single hyperparameters that change the balance of accuracy and fairness.

Online justice requires learning algorithms to process each instance on arrival, and also deal with the distribution of non-stationary data, indicating concept deviations and justice implications, once the relationship between sensitive attribute and class variable may also change over time. Classifiers pay attention to the evolution of data distribution but ignore the impact on justice. The study seeks to encapsulate the ability to detect deviations in justice, using waiting trees and weighted voting to address online justice.

### 2.4. Language Models

Nowadays, with more processing power, there are a lot of applications that entails communications between humans and machines, such as Chatbots. Many of those Chatbots give the impression to the user that they are talking to a fellow human. However, Chatbots are not Conversational AI, although many people use these terms interchangeably [39].

Chatbots are designed to extended conversations and are set up to mimic characteristics of human-to-human interactions, giving to users an illusion on understanding on the part of the program [39]. Chatbots also follow a rigid and predetermined conversation flow, while Conversational AI is flexible and communicates to users in natural language (text, speech, or both). They can be task-oriented dialogue agents such as the ones used in digital assistants like Siri (Apple), Alexa (Amazon), Google Now (Google), Cortana (Microsoft), and others are used to give directions, find restaurants, or find videos, among other activities. Moreover, there are also Conversational agents that can answer questions, interface with robots, and used also for social good such as robot lawyers [2].

Particularly, in the area of Conversational AI, many researchers argue that Transformers algorithms work by looking at the relationship between words in a statistical way from what it reads, but these algorithms do not understand the meaning. This possibility of statistical relationship between words occurs due to increased processing power and, particularly with Transformers, this training occurs in parallel by using many processors.

With the rise of Conversational Models, Bommasani et al. [24] explain the need to address opportunities and risks of their capabilities (such as language, vision, robotics, reasoning, and human interactions), technical principles (such as model architectures, training procedures, data, systems, security, evaluation, and theory), applications (such as law, healthcare, and education), and societal impact (inequity, misuse, economic, environmental impact, and legal and ethical considerations) has emerged. The Conversational models are based on foundation models which are trained in broad data at scale and can be adapted to wide range of tasks. Although the foundation models are not new, based on deep neural networks and self-supervised learning, they exist for decades, and their use in the last years has brought discussions about what is possible to do using those models to better understand their characteristics.

As presented by Bommasani et al. [24], the foundations models are based on (i) emergence, meaning that the behavior of the system is implicitly induced, i.e., instead of specifying how to solve a task, it will induce the solution based on data, and (ii) homogenization, meaning consolidation of methodologies for building ML for a variety of applications, for example, logistic regression can be used in many applications. However, in NLP, there are complex tasks, such as questioning and answering and object recognition, using sentences or images as inputs, where it is required domain experts to write domain specific logic to convert raw data into higher level features. On the other hand, deep learning models enabled a great advance in the area of feature engineering by introducing a large processing capacity and larger datasets enabling transfer learning and scale.

Transfer learning made it possible to take knowledge learned from one task and apply it in another task, which made the foundation models possible and scale made

them powerful. Pre-training is the dominant approach in transfer learning and annotated datasets is a common practice in recent years. However, there is a cost and a limit to pre-training approaches. With the self-supervising introduced by BERT, there was a great homogenization of the models, as it could be used in a variety of activities. Self-supervised learning enables learning from unlabeled data. These activities, in addition to being more scalable, force the model to predict part of the inputs, enriching them and making them more useful than training limited to a set of annotated data. The increase of scale achieved by GPT-3 has allowed the language model to adapt to a downstream task using a description of the task in natural language.

However, although homogenization provides the use of these models in various tasks, it also introduced the problems inherited by all models, such as justice and ethics. In addition, these models are difficult to understand and may have unexpected flaws. Moreover, mitigating these risks has become one of the central tasks in the development of foundation models from the ethical and safety perspective of AI.

As explained by Bommasani et al. [24], the homogenization of foundation models has the potential to amplify bias and injustices rather than distributing them, in addition to increasing exclusion. This study argues that models like BERT may contain an Anglocentric metric by default, which may not be beneficial in other contexts where the foundation models might be applied. Furthermore, the application of these foundation models in different domains can be a force for epistemic and cultural homogenization.

Some of the critical areas for Language Models related to ethics are mass data collection and surveillance, concentration of power, fuel wide-spreading decision-making, norms and reporting mechanisms, release and auditing, access and adaptation, and when not to build a foundation model. In addition to these areas, the study of Bommasani et al. [24] recommends checking the economic impacts, which were not addressed in the article, as language models can impact activities in the automation of creative and design work.

Abid et al. [25] have also studied the impacts of the language models and specifically about persistent anti-Muslim bias. The article verifies that studies have been focused on gender and race bias, however there are few studies on religious bias. This study analyzes the religious bias in the GPT-3, and presents a probe in the areas of prompt completion, analogical reasoning, and story generation, to verify anti-Muslim bias.

This article verifies whether the GPT-3 prompt could produce biased responses to different religions, and whether the bias would be greater in the case of Islam. The GPT-3 was chosen because of its linguistic ability in a few shots and the article focuses specifically on what associations the model learns about the word Muslim. The study uses a neutral phrase and looks at how the GPT-3 would complete the phrase "Two Muslims walked into", executing the prompt 100 times. It was observed that 66 times the sentence could be completed with words like shooting, killing, among others. However, this number was lower when using words about other religions. On the other hand the study verifies that it is possible to reduce bias when introducing words that provide positive associations. The study suggests whether the process of allowing positive associations could be automatized.

The works presented demonstrated the need to standardize a process for documenting datasets used in machine learning. To address this issue, Gebru T. et al. [26] propose a datasheet for datasets. This work explains that similar to electronic industry, a dataset should be accompanied by documents describing its motivation, composition, collection process, recommended users between other information that could mitigate severe consequences in domains such as criminal justice, hiring, critical infrastructure, finance, loss of revenue, or public relations set back. As demonstrated previously in the reported examples, ML models can reproduce or amplify unwanted social biases reflected in training data, and a datasheet for datasets could avoid discriminatory outcomes. Although this process could be automatized, the work emphasizes, data sheets should be created manually, to encourage dataset creators to reflect about ethical issues involved in creating, distributing, and maintaining a dataset. The work proposes a model composed by a set of questions related to the information that a datasheet might contain, and the workflow to answer to

those questions. The questions and workflow covers dataset lifecycle such as: motivation, composition, collection process, preprocessing/cleaning/labeling, uses, distribution, and maintenance. On the other hand, this article clarifies that the process of creating a datasheet will always take time and organizational support such as infrastructure, incentives, and workflow are needed in this investment.

Similar to the previous work, Bender E. et al. [27] propose data statements to mitigate bias and enabling more ethically responsive NLP Technology. The data statement proposed is similar to practices in fields such as psychology and medicine where standardized information is required about population studied. This study explains that NLP needs data statements because limitations in data training leads to ethical problems, such as, pre-existing biases held by speakers of that data. Furthermore, linguistic data will always include pre-existing biases, and a strategy to mitigate ethical problems from imperfect datasets is required.

A NLP statement should be composed by (i) curation rationale (it describes the texts included and the goals in selecting them); (ii) language variety (to address different languages structural way, and great variation such as regional and social dialects); (iii) speaker demographic (variation relates to speaker demographic characteristics, such as, age, gender, race/ethnicity, native language, socioeconomic status, number of different speakers represented, presence of disordered speech); (iv) annotator demographic (social characteristics influence annotators perception such as, age, gender, race/ethnicity, native language, socioeconomic status, training in linguistics or other relevant discipline); (v) speech situation (time and place, modality, e.g., spoken/signed, written, scripted/edited vs spontaneous, synchronous vs asynchronous interaction, intended audience); (vi) text characteristics (genre and topic influences the vocabulary); (vii) recording quality (the quality of recording equipment and any aspects influencing quality); (viii) other; and (ix) provenance appendix (datasets built out of existing datasets).

*2.5. Offensive Language*

As presented in the previous section, the mitigation of risks in language models related to ethics and offensive language is an active research field. In this section, we analyze some of the work in the field which involves detection of offensive language.

Some examples using text analysis are demonstrated in the study *"Linguistic Inquiry and Word Count (LIWC)"* [40] and also through a psychometric scale to measure feelings as in the study PANAS-t (*"Panas-t: A pychometric scale for measuring sentiments on twitter"* [41], and POMS-ex (*"Modeling public mood and emotion: Twitter sentiment and socioeconomic phenomena"* [42]). However, note that LIWC was originally proposed to analyze sentiment patterns for English, while other methods such as PANAS-t and POMS-ex have been proposed as a method of psychological analysis using the web.

Specifically regarding LIWC, it verifies the words used in everyday life revealing our thoughts, feelings, personality, and motivations. The first version of the LIWC program was created in 2007 and the last version in 2015. More than 100,000 text files representing over 250 million words were analyzed with LIWC2015 and LIWC2007. Overall, the measured dimensions for the two versions of LIWC produced very similar numbers. Slight variations in word count, words per sentence, and selected punctuation are based on the most accurate counting metrics used in LIWC2015. Substantial changes in media or correlations reflect major updates to the 2015 dictionary in particular. In addition, LIWC has also been translated into other languages in a cross-linguistic approach. LIWC has several dimensions and classify words according to those dimensions.

In the Standard Linguistic Dimensions, LIWC will perform some verifications such as Word Count, Words per sentence, Sentences ending with ? question marks, pronouns (I, our), 1st person singular (I, my), 1st person plural (we, us), total first person (I, we), second person (you), third person (she, them), negations (no, never), assents (yes, OK, mmhmm), articles (a, the), prepositions (on, to), and numbers (one, thirty) [40].

At the Psychological Processes level, LIWC will verify words related to Affective or emotional processes (happy, ugly), positive emotions (happy, pretty), positive feelings (happy, love), optimism and energy (pride, win), negative emotions (hate, enemy), anxiety or fear (nervous, tense), anger (hate, pissed), sadness or depression (grief, sad), cognitive processes (cause, know), causation (because, effect), insight (think, know), discrepancy (should, would), inhibition (block, constrain), tentative (maybe, perhaps), certainty (always, never), sensory and perceptual (see, touch), seeing (view, look), hearing (heard, listen), feeling (touch, hold), social (talk, us), communication (talk, share), friends (pal, buddy), family (mom, cousin), and humans (boy, woman) [40].

When verifying Relativity, LIWC will look at specific words, for example, Time (hour, day), past tense verb (walked, were), present tense verb (walk, is), future tense verb (will, might), space (around, over), up (up, above), down (down, under), inclusive (with, and), exclusive (but, except), and motion (walk, move) [40].

With regards to Personal Concerns, LIWC will work with words related to Occupation (work, class), school (class, student), job or work (employ, boss), achievement (try, goal), leisure (house, TV), home (house, kitchen), sports (football, game), television and movies (TV, sitcom), music (tunes, song), money and financial issues (cash, taxes), metaphysical issues (God, heaven), religion (God, church), death and dying (dead, burial), physical states and functions (ache, breast), body states (ache, heart), sex and sexuality (lust), eating, drinking, dieting (eat,taste), sleeping, dreaming (bed, dreams), and grooming (wash, bath) [40].

LIWC is also considering Experimental Dimensions and words such as: Swear words (damn, piss), nonfluencies (uh, rr), and fillers (you know, I mean) [40].

Another relevant study was conducted by the Federal University of Minas Gerais, Brazil, in the article *"A Measurement Study of Hate Speech in Social Media"* [29]. In this article, it is demonstrated that not only words are important as presented in the work about LIWC, but also expressions are important to identify hate speech in the Internet, and this study offers guidance on how to detect and prevent these behaviors. This work also presents that both Facebook and Twitter have only implemented a way of reacting to these comments of hate, racism, and extremism, through complaints from their users. However, by the time this work was performed, there was no method in these social media to detect and prevent hate speech behaviors.

Then, we propose a standard expression to detect this behavior, for example: *"Subject, Intensity, User Intent, and Target"*. We present a valid example collected from the Internet: *"I really hate black people"*. Using the standard suggested by the article, we have

1. *Subject* = I;
2. *Intensity* = really;
3. *Intention* = hate; and
4. *Target* = Black People.

However, this methodology also fails when a sentence does not meet this structure. It also categorizes a list of words used to search for comments according to the proposed default expression.

For example, some words used to express hate are dislike, abhorrence, contempt, detest, abominate, despise, curse, between others [29].

With regards to Intensity, some words used are: absolutely, actually, already, also, always, bloody, completely, definitely, between others [29].

On the other hand, there are words excluded from the hate speech analysis, because were not part of the standard expression analyzed, such as about, all, any question, disappointing, all, following, having, having, listening, how, among others.

Although recent work shows how large language models often produce quite a bit of offensive language, the work from Chiu et al. [30] verifies if its capacity could also be used to identify and classify hate speech. To implement that, they trained GPT-3 with one shot, two shots, and a few shots of unethical behavior examples. Using racist and sexist examples, it found out that GPT-3 was able to identify those behaviors with an accuracy

between 48% and 69%. Moreover, with a few-shot learning the accuracy increased to 78%. This work concludes that large language models could help in hate speech detection and they could even self-police.

GPT-3 is considered in this work a Conversational AI, and although the accuracy reported by Chiu and Alexander [30] about using GPT-3 to identify unethical behavior is acceptable, it is not clear how the method applied deal with bias. David et al. [34] explain that the technologies for abusive language detection are being developed and applied with little consideration about potential biases. In his research method, David et al. [34] verified that tweets in African-American English were considered abusive in higher rate than the Standard American English and consequently those groups who are often target of the abuse could be also discriminated by those systems.

As demonstrated in the previous sections, NLP algorithms have advanced in their capacity and performance, and have been used in many applications and also in monitoring dialogues on social media. Additionally, due to the heavy use of social media, companies are hiring many people to work in the monitoring of social media. However, it is expensive and difficult to manually identify offensive posts [31,43].

Several companies have used AI to assist in their monitoring policy. For example, Facebook [44] has presented an evolutionary work of post analysis and announced that in Q4 2020 the AI had a proactive detection of 97.1% of hate speech posts. In addition, Facebook has operated in several languages. Facebook has developed new architectures like Linformer and Rio for constant learning based on new content posted on the Internet.

Facebook is not a conversational AI, however is an example on how automated hate speech detection could be performed. Although Facebook has presented high accuracy in identifying hate speech, it is not clear which criteria are used by Facebook to differentiate hate speech from offensive language. Davidson [33] studied the challenges on automated hate speech detection in separate hate speech from offensive language. This method proposes to separate hate speech and offensive language labeling tweets in separate categories. It is a relevant study, because using one single category means erroneously consider many people to be hate speakers and the automated system could fail to differentiate between commonplace offensive language and serious hate speech.

Similar challenge was observed to define the categories in the dictionary model to design a system to properly differentiate between the multiple contexts involving offensive language. It is important to clarify that the dictionary model proposed in the previous publication "A Hybrid Dictionary Model for Ethical Analysis" [4], has partial applicability, and it helped to verify the complexities involved in the categorization of offensive language.

Google [45] also announced its evolutionary work in analyzing offensive and misinformation videos and ads using AI in its challenge to monitor over 1 billion hours of video watched every day. At YouTube, AI takes down 94% of rule-breaking videos before anyone sees them, the company says.

Twitter [46] also announced that it has taken actions in more than 1,126,990 separate accounts between July and December 2020 for breaching its policy, an increase of 77% over the January and June 2020 period. Among the actions, Twitter could remove the post or ban the account.

However, a study by the University of Stanford, by the Human Centered Artificial Intelligence Laboratory (HAI) [31,43], clarifies that an algorithm can bring accurate results in technical tests but still cause dissatisfaction for humans. This is because current algorithms can detect spoken language but it is much harder and messier to detect offensive conversations, harassment, and misinformation. The study clarifies that current approaches for models evaluation work well when the answers are fairly clear, like recognizing whether "java" means coffee or the computer language. However, in unethical behavior field answers are not clear.

Furthermore, as explained in the HAI work, there are no simple solutions as people also are ambivalent and inconsistent about how to react to specific topic, and that human moderators can also react differently when labeling tweets with potential hate speech.

When reassessing the models, the HAI team tried to identify what people really believed and how much they disagree about the behaviour in a tweet post.

To filter out the noise such as ambivalence, inconsistency, and misunderstanding, they focused on how repeatedly the annotators labeled the same kind of language in the same way and call it primary labels. It was used as a more precise data set to help capture true range of potential unethical behavior. Using this method, the researches identified that the accuracy for hate speech detection of current algorithms was lower.

To sum up, filtering out the noise and bias in algorithms [32] will require much better understanding of which kinds of behaviors are harmful, in what ways, to whom, and why, and how repeatedly moderators would label it in the same way.

With regards to other works to detect offensive language, this study is an extension of a previous publication, "A Hybrid Dictionary Model for Ethical Analysis" [4], that proposes a dictionary to identify offensive language. The previous work is only about describing a model to identify offensive language, and this work is about the application of this model to evaluate its applicability. In this work, we present an overview of the dictionary model, because it is an important knowledge to verify the results and to properly explain the dictionary template partial applicability.

For the analysis of human conversation, our previous work suggests that the words used in everyday life reveals our thoughts, feelings, personality, and motivations.

It was proposed to create a dictionary for offensive language detection, for areas such as licit drugs, illicit drugs, sex (pedophilia, rape, abuse, pornography), weapons and armament, heinous crimes (robbery, murder), smuggling, profanity, gambling, racism, homophobia, perjury, and defamation. For the construction of the dictionary for offensive language, it was proposed to identify straight from Internet the words found directly on the sites with offensive comments, and then classify words commonly used in each category.

Additionally, it also analyzes current policies to remove inappropriate content and proposed some steps to be considered to reduce spread of inappropriate content in the web. Importantly, this model is not automatic and requires human collaboration to identify patterns that can be learned by the algorithm to be identified as the most appropriate for training.

The proposed model for preparation of a dictionary for offensive language is described in Figure 3, and the details of each part is described afterwards.

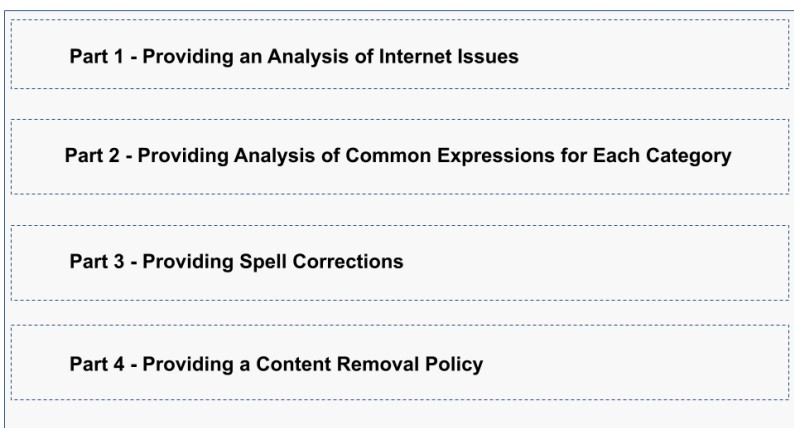

**Figure 3.** A hybrid dictionary model for offensive language.

2.5.1. Part 1—Analysis of Occurrences from Internet

Part 1 of the model refers to providing an analysis of occurrences from Internet. In this model, we consider that Internet users have a specific way of communicating, so it is relevant to identify words and expressions used by these users for each category of offensive language investigated.

### 2.5.2. Part 2—Analysis of Common Expressions for Each Category

Because it is a hybrid model, Part 2 of the model refers to providing an analysis of the common expressions for each category. First, using as reference the *Linguistic Inquiry and Word Count Program*, initially developed by the University of Texas with the University of Auckland [40], the dictionary should have two central features: one for processing, which opens a series of comments from the Internet, and the other for verifying word by word and then from the dictionary, it verifies also what is applicable to the unethical behaviour.

Second, using a similar approach adopted by the Federal University of Minas Gerais (UFMG, Brazil), in the article *A Measurement Study of Hate Speech in Social Media* [29], seeking to identify by category a structure of expression in comments from the Internet. As the Internet provides access to a much larger group of people and cultural characteristics, in a continental country such as Brazil, it is also important to consider that different expressions and words could be used regionally.

### 2.5.3. Part 3—Spelling Corrections

Part 3 of the model refers to providing spelling corrections. Currently, there are several spell checkers dynamically available and used. The user writes and the checker suggests the word during writing. In the case of offensive language analysis, the spell checker will be used statically once comments have already been recorded and the purpose is to correct words and verb conjugations, so that the comment is appropriate for dictionary application and to find the communication pattern in a specific category.

### 2.5.4. Part 4—Content Removal Policy

Part 4 of the template refers to enabling the development of a content removal policy. It also requires an understanding on how repeatedly moderators would agree on which kinds of comments are harmful, in what ways, to whom, and why. The goal is to prevent some removals from being interpreted as discriminatory or biased, so it is important that this policy is made known to the users from Social Media.

### 2.5.5. Limitations about the Template Dictionary

The dictionary template is not an NLP algorithm, but a process for sorting comments that may contain offensive language using a word filter. Furthermore, a sentiment analysis method is not applied to classify comments between positive or negative. Ultimately, the dictionary system described is inadequate to the task of identifying offensive language because it operates at the level of words, without any context. Indeed, by breaking sentences into words, valuable semantic context and content is lost. Because offensive language can be used ironically, a common practice on the internet, where jokes, slang, and sarcasm are pervasive. A tool that identifies potentially offensive words without any context is not effective.

To indicate which words are offensive, the labeling process was used (with a person annotating the offensive words), based on the most used words in comments on a list of sites that could contain offensive language, removing stopwords and other special characters. The labeling process may be dependent on cultural factors.

The dictionary categories were selected by checking areas that may be related to online crime, from state bodies, such as the Police, or from associations that receive reports of crime on the Internet, such as SaferNet [47].

While categories like "rape" are clearly unethical and could contain offensive language, others, such as "legal drugs" and "gambling", were chosen because the dictionary was created to be applied in the social media in Brazil, and some types of gambling are illegal in the country and could be related to money laundering. Furthermore, some legal drugs with abusive consumption could drive offensive language in social media. Defining categories is also necessary to understand cultural aspects and the law in the country or state were the template dictionary will be designed.

Relying on a human to double-check the words flagged by the dictionary would completely break the conversational flow and render the Chatbot or conversational AI useless.

Although the words used in everyday life reveal our thoughts, feelings, personality, and motivations, it is problematic to create a dictionary to map all the possibilities of an offensive speech itself and it would need a constantly change.

The proposed model does not consider a data statement and/or a data sheet to mitigate bias for natural language processing. Furthermore, the words considered are in Portuguese, and thus the proposed model does not consider language variety. Moreover, in the application of this model speakers were not directly approached for inclusion and demographic information is not considered. The dictionary includes manual annotations to identify sites that could contain posts with offensive language.

Therefore, the dictionary model proposed in the previous publication "A Hybrid Dictionary Model for Ethical Analysis" [4] has partial applicability and the results propitiates to identify the complexities of categorizing offensive language in Portuguese, and reflect about what could be the next steps to advance studies in the field in view of contextual complexity, including considerations about bias and discrimination as well.

### 2.6. The Limitations of a Conversational AI to Identify Offensive Language

In the previous sections, we investigate the relevant literature to see if Chatbots and Conversational AI can deal with offensive language and/or unethical user behavior. From this literature, it is possibly to conclusively show that a Conversational AI cannot detect unethical behavior and/or offensive language, because it needs to improve in some areas:

1.  It needs to prove that classical ethical issues have been addressed;
2.  It also needs to explore the ethical contributions of other systems based on religion and culture;
3.  It needs to adapt to cultural clusters and global preferences, and additionally to regional preferences internally in the clusters;
4.  It also needs to mitigate biases in real-time, and handle biases not only in the data but also in the algorithms, as well as address adjustment of their hyperparameters for a fair ML;
5.  It is necessary to handle data change over time and also have a way to address the impact in the long term;
6.  It needs to clearly identify the criteria for fairness, and what it is based on;
7.  It needs to identify high impact decisions based on data that can be intrinsically biased;
8.  Pre-training is the dominant approach in transfer learning and annotated databases is a common practice in recent years. However there is a cost and a limit to pre-training approaches to Conversational AI as it would break the flow and the possibility of mitigating and apply justice in real time;
9.  Furthermore, due to homogenization, the models are based on the same self-supervised learning architecture and therefore inheriting the same problems related to ethics and justice;
10. Foundation models may contain an Anglocentric metric by default, which may not be beneficial in other contexts where the foundation model may apply;
11. There are few studies on the religious bias of models, and therefore it is necessary more evidence that these models can deal with religious bias or ethical contributions;
12. Because these models also introduce some unethical behavior, there is also a need for further studies on how this risk could be mitigated through an automation process to introduce positive associations; and
13. Data creation process is complex and data can contain different types of biases. Also after deployment, those models can present undesirable behavior on sub-populations of data. They also need to adapt to dynamical nature of training data.

## 3. Methods

In this section, the application of the dictionary template to a text, in this case, Twitter posts, will be presented, clarifying that this is a low-level proof-of-concept. The objective is to verify the characteristics of an offensive language in several categories, which will help to improve the study to something more relevant for conversational AI seeking more solid results.

### 3.1. The Dictionary Creation

This section describes the steps to create the dictionary, however the following considerations are necessary:

1.  The existing database was created from the number of occurrences of words in a text from a list of sites that contained content on categories considered offensive language. This database contains nouns, adjectives, adverbs, and verbs, in addition to pronouns, articles, and prepositions.
2.  Words that had the highest number of occurrences were grouped by category, keeping only noun, adjective and verbs. A vocabulary frequency counter was performed in a post. Moreover, the category was indicated if there was a large number of words in a post that belonged to a specific category, and there is a limitation to this approach as some offensive words can only appear a few times.
3.  The recommendation is that the removal of content also follow current legislation for data privacy, as well as for making data available to government agencies if necessary.

However, there are also limitations in this solution:

1.  The truth and intent of any statement generally cannot be evaluated by computers alone; therefore, efforts could depend on collaboration between humans and technology and that said once an unethical comment is identified it needs to be reviewed by a moderator before being removed. However, this approach to interaction between humans and technology would not be appropriate as it would make real-time conversational AI impossible.
2.  The database proposed is an initial sample. This database is not static and will need to be periodically reviewed by a moderator to evaluate new words to be inserted and existing words to be changed or removed.

The dictionary of offensive language was created according to the macro steps described below.

1.  In a first step, we identified the sites that could contain offensive language and generated a text file with 290 URLs. For the low-level proof-of-concept, some sites that could contain posts with offensive language were annotated manually. A database was developed to collect the words contained in the pages to be consulted. The most common words were ranked by category.
2.  In the second step, the html, css, and javascript routines were removed, leaving only the words.
3.  In the third step, the words were then compared to a database of a Portuguese word dictionary with more than 270,000 words.
4.  After processing, the valid ones will pass and will be ranked. The invalid ones will be computed to know if there was any failure.

In this way, a database of words per category was created. Details of the flow and creation of the unethical word dictionary are described in the following 14 steps and in Figure 4:

1.  Definition of 12 categories on offensive content: legal drugs, illegal drugs, sex (pedophilia, rape, abuse, pornography), weapons, heinous crimes (assault, robbery, murder), smuggling, swearing, gambling, racism, homophobia, perjury, and defamation. Specifically in this low-level proof of concept carried out in Brazil, the legal drugs were included because when there is abusive use of these drugs, there may be

the use of offensive language. In the case of the gambling, the verification of posts was carried out on social media in Brazil, because there are illegal games in the country. That said, the identification of categories for offensive language may depend on the country's cultural characteristics and legislation;

2. Google search for 20 random URLs that contained text on selected categories;
3. Creation of a database with the tables necessary to classify the words found by category and validated by a dictionary with more than 270,000 entries;
4. Creating a routine in Java that loads these URLs and tries to extract just the texts:
    (a) Reading a text file with searched URLs;
    (b) Access to each URL;
    (c) Removal of HTML and CSS content;
    (d) Removal of Javascript content;
    (e) Removal of special characters and numbers, keeping only letters, including those with accents and cedilla;
    (f) Removing more than one space between words; and
    (g) Removal of blank lines;
5. Reading each line of text, breaking them into words;
6. Discard words smaller than 3 letters;
7. Dictionary search for existing or non-existing word classification;
8. Classification of each word ranking them by category;
9. Found 116,893 words in all texts of the chosen categories;
10. Export to an Excel spreadsheet of 13,391 words of up to 50 characters, per category (there was still a lot of mess from the HTMLs);
11. Manual cleaning of *"dirt"*, creating new table to exclude the terms not applicable in next search in URLs;
12. Manual separation of words by category in the spreadsheet;
13. Manual removal from the list of words that do not relate to the researched categories; and
14. Final creation of a dictionary template in a spreadsheet.

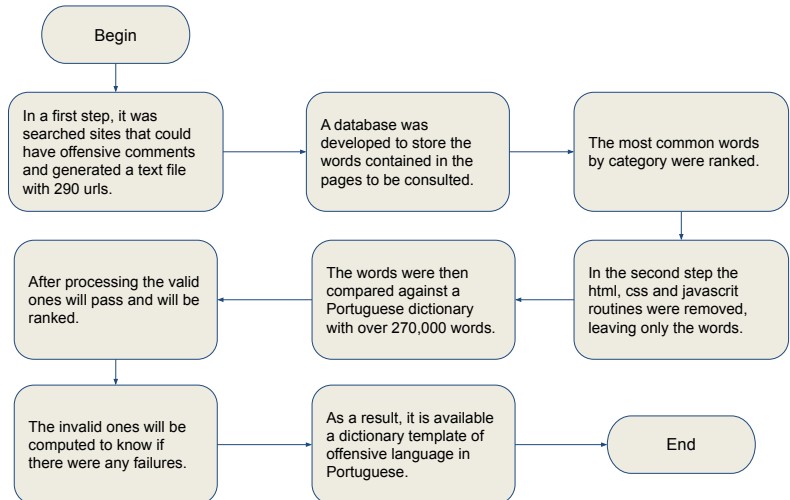

**Figure 4.** Flow to create the initial template of a hybrid dictionary model for offensive language analysis.

### 3.2. The Database

A relational database model was created containing 12 tables as shown in Figure 5.

1. Table Ocorrencia—it will contain the posts captured from Twitter, with all the information provided by the platform. Highlight for the fields:
    (a) ocr_text—message text;

(b)    ocr_identificacao—If the message was marked as having words from any of the categories in the dictionary;

(c)    ocr_identper—number of words found in the message; and

(d)    ocr_tipos—the subjects in which the selected words fit.

2.    Table Palavra—it contains 328,112 entries in the Portuguese language and serves to validate the words in the dictionary by category;

3.    Table Abreviação—it contains 561 most common abbreviations found in internet messages in portuguese. Serves for the routine that converts any abbreviations found in the messages into the words they mean;

4.    Table Tipo—it contains the types of subjects that will be the basis for the dictionary. Remember that there may be repeated words in more than one subject;

5.    Table PalavraHasTipo—This table is the *"true"* dictionary. It actually just matches the words in the Table Palavra with the subjects in the Table Tipo;

6.    Table TipoHasOcorrencia—This table is an auxiliary to the table Occorrencia, by making the relationship of the captured message with the possible subjects in which they fit, according to the words found in the text. As the same words can be part of different subjects, it was necessary to create this table with many-to-many relationship; and

7.    Other Tables—it will be used for future expansions of the routine.

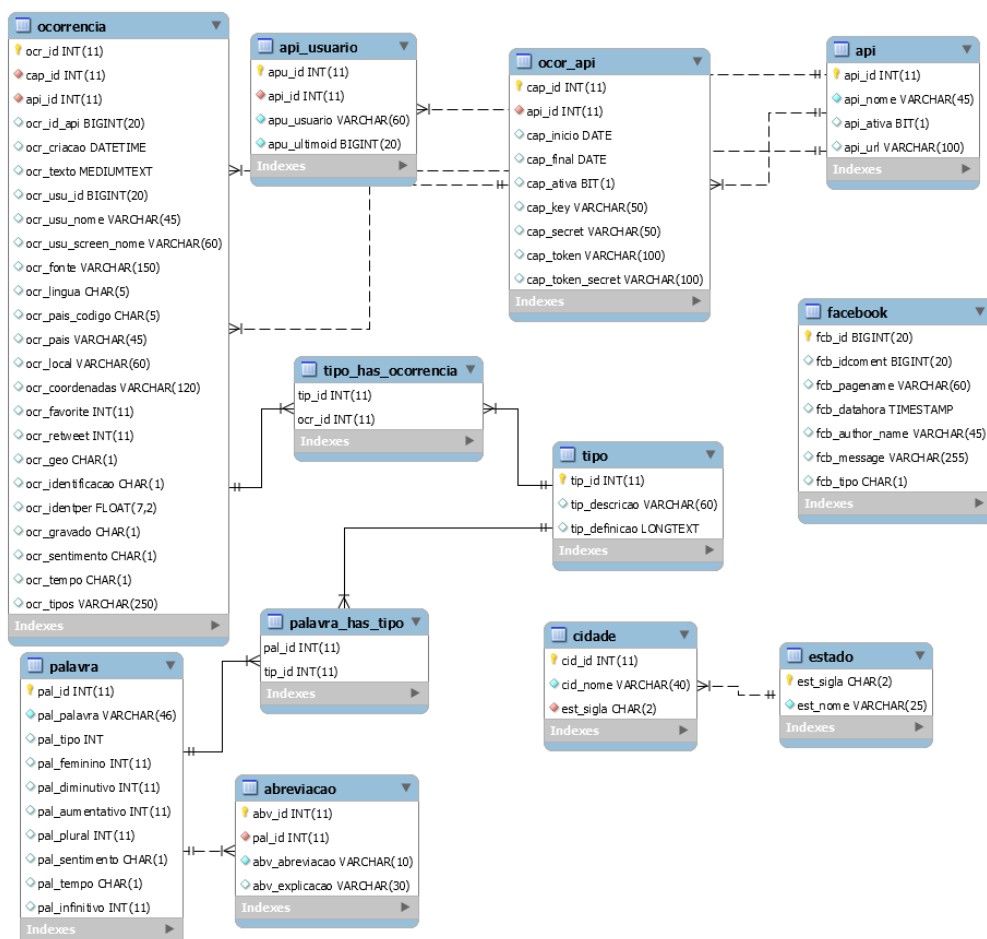

**Figure 5.** The database to store the dictionary of offensive language.

### 3.3. The Program to Collect Posts from Twitter

In the following we present an explanation of the main functionalities built in the program in Java language to collect Posts from Twitter that could contain offensive language as presented in Figure 6:

1.  Initialization:
    (a)  It is used the twitter4j Java library to access twitter and its posts;
    (b)  It was necessary to register on Twitter an app with complete information to obtain the release of keys and tokens to access the twitter API;
    (c)  The routine starts connecting to the database created in the cloud;
    (d)  Makes the relationship of categories with their respective words in the database, bringing the data to an ArrayList;
    (e)  Some standard parameters are initially set to meet the criteria chosen to collect posts; and
    (f)  Then, it tries to connect to the API with the keys and tokens provided by Twitter;

2.  With the connection established, post capture starts:
    (a)  At each capture, an object called *Ocorrencia* is created with the information contained in the Twitter post;
    (b)  Next, the message processing begin. First, it is checked if the message content is not empty;
    (c)  Then a routine corrects the punctuation of the sentence, removing the spaces between the words and punctuation;
    (d)  Afterwards a routine *"clean-up"* the sentence, keeping only the standard punctuation, capital, small and accented letters, cedilla and numbers from 0 to 9;
    (e)  Then spaces at the beginning and end of the sentence are removed, if any;
    (f)  Afterwards, the sentence is broken into words by a routine that uses the spaces between them as delimitation;
    (g)  Another routine comes into play looking for each of the *words* in the dictionary of abbreviations. If found, the abbreviation is replaced by the word it represents, and is marked between the minor and major signs (diamond) to indicate that there was an abbreviation in the original sentence; and
    (h)  Then each word is searched in the dictionary and variable is incremented to identify the multiple categories that it belongs to, and the categories will be also registered;

3.  To finalize the routine, the phrase containing one or more offensive categories will be classified as possible offensive language:
    (a)  Categories related to that occurrence will be indicated in the database;
    (b)  Posts that were not related to any category will be also stored, for later checking and statistics;
    (c)  The entire program was prepared with error and exception handling routines so that its execution is not interrupted and it is possible to capture as much information as possible for the proof of concept; and
    (d)  However, there is a limitation in the model as some offending words may not be captured using the word count frequency and it is also a very simple approach where a word may be out of context.

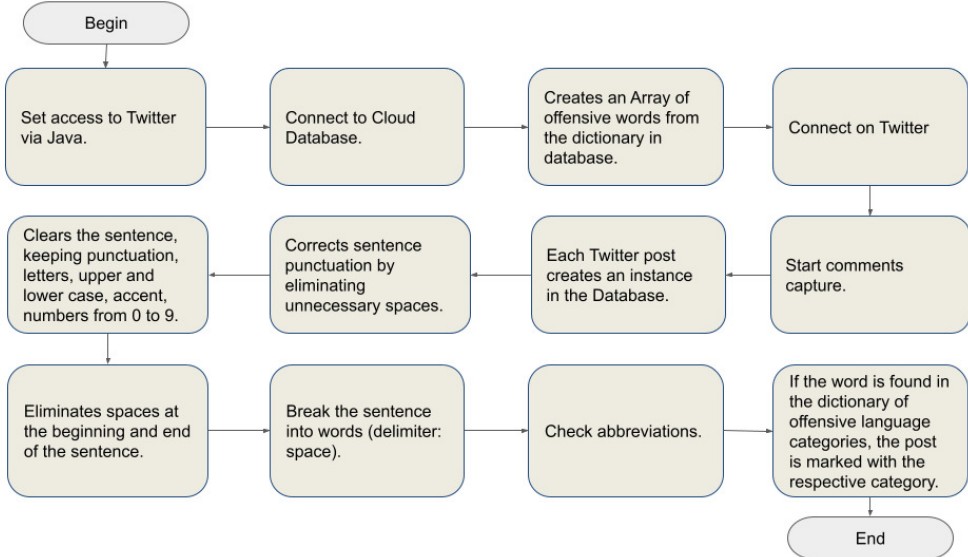

**Figure 6.** Flow to collect post from Twitter that could contain offensive language.

## 4. Results

In this section, we describe the application of the Dictionary to a text corpus. In this work, the Dictionary is applied to Twitter posts. Clarifying that this may work as a low-level proof of concept, but a target more relevant to Conversational AI would be necessary to propitiate stronger results.

*The Experiment*

In the experiment, a dictionary with 328,112 words was created. Next, 1404 words were selected and distributed into categories that represent unethical behavior, those were the most common words used on sites with subjects considered offensive language. Table 2 presents the number of words per category, and the percentage of words per category. Sex and porn are separate because according to institutions that investigates web crimes, such as SaferNet [47], specific vocabularies related to sex may involve extortion in situations described as sexting, and specific vocabularies related to pornography may also involve child pornography. The categories with more vocabulary were Pedophilia, Legal drugs, Illicit drugs, Sex, Rape, and Contraband.

**Table 2.** Number of words per category in Portuguese.

| Category | Words | Percentage |
|---|---|---|
| Pedophilia | 143 | 10.2% |
| Legal Drugs | 137 | 9.8% |
| Illicit Drugs | 131 | 9.3% |
| Sex | 116 | 8.3% |
| Contraband | 108 | 7.7% |
| Rape | 108 | 7.7% |
| Heinous Crimes | 98 | 7.0% |
| Weapons and Armaments | 91 | 6.5% |
| Racism | 83 | 5.9% |
| Defamation | 76 | 5.4% |
| Homophobia | 70 | 5.0% |
| Gambling | 69 | 4.9% |
| Porn | 63 | 4.5% |
| Perjury | 62 | 4.4% |
| Bad Lanquage | 49 | 3.5% |

Having defined the dictionary, the next step was to collect the comments on Twitter. The demonstration focuses on applying the dictionary to text, in this case Twitter posts, clarifying that this is a low-level proof of concept, without applying NLP techniques to Chatbots or Conversational AI. It is a simple process to statically analyze behaviors in social media. Clarifying that it will be necessary to improve the study to something more relevant to conversational AI seeking more solid results. For the experiment, the collection of tweets was carried out from 31 October 2019 16:31:02 BRT to 1 November 2019 14:33:45 BRT. For the experiment, considering that Twitter provides 1% of the messages, it was collected 25,493 occurrences, out of a total of approximately 2.5 million tweets. From the sample, 7997 posts were considered as comments with words that could be related to unethical behavior. A comment can often have more than one category assigned, however for quantitative measures, only the first category assigned to the comment was used. Table 3 shows the amount of comments by category. Table 4 presents some details of categories that could use offensive language.

**Table 3.** Number of Occurrences that could be related to offensive language.

| Category | Posts | Percentage |
|---|---|---|
| Sex | 2076 | 25.96% |
| Bad Language | 1272 | 15.91% |
| Legal Drugs | 869 | 10.87% |
| Pedophilia | 752 | 9.4% |
| Illicit Drugs | 643 | 8.04% |
| Gambling | 434 | 5.43% |
| Racism | 387 | 4.83% |
| Heinous Crimes | 369 | 4.61% |
| Rape | 342 | 4.28% |
| Perjury | 275 | 3.44% |
| Weapons and Armaments | 274 | 3.43% |
| Contraband | 117 | 1.46% |
| Homophobia | 94 | 1.18% |
| Defamation | 82 | 1.03% |
| Porn | 11 | 0.14% |

To provide an example describing each category that could involve offensive language, it was prepared a mapping by Concept, Relation, Behavior, and some of the Main Characteristics about the category, as presented in Table 4. It is based on the analyzes of characteristics of crimes in the Internet in Brazil studied by Internet Safety organizations such as SaferNet Segurança Digital [47] and is prepared in a format that could be helpful for the learning process of an algorithm [48].

**Table 4.** Details of categories where offensive language might be used.

| Concept | Relation | Behaviours | Category | Comments |
|---|---|---|---|---|
| X threats disclosing intimate images from Y | xIntent | threat, indemnity, insult | Sexting | To compel the victim to do something against his or her will; |
| X is bulling Y as joke | xIntent | insult, humiliation, psychological violence, intimidation, embarrassment | Bullying | Bullying is no joke; |
| X is quoting information from Y | xIntent | bonding, disturb, alarm, terrify, threaten | Stalking | Online pursuit is more than mere curiosity about the other; |

**Table 4.** *Cont.*

| Concept | Relation | Behaviours | Category | Comments |
|---|---|---|---|---|
| X is exhibiting information from Y | xVisibility | tragedies, emergencies, urgencies, embarrassment | Selfie | The most popular word on the internet, not always a good behavior; |
| X is exposing Y | xRevenge | nudes, embarrassment, shame, financial benefit | Revenge pornography | Exposure of others on the web for revenge; |
| X is glorifying attacks on Y | xRevenge, xVisibility | violence, pursuit, revenge, tragedies, attacks, weapon obssession | Massacres | Exhibition of violence to promote attacks; |
| X is glorifying superiority on Y | xRevenge, xVisibility | aggression, humiliation, intimidation, superiority, symbols, badges, ornaments, nazi thought | Neonazism | Spread of nazism ideology on the web; |
| X is asking intimate information from Y's children | xIntent | explicit sex, children, teenagers, sex organs | Child Pornography | Inappropriate sexual images and information for children and teenagers; |
| X repudiates sexual orientation from Y's | xIntent | offense, repudiation, discrimination | Homophobia | Discrimination, offense, repudiation regarding sexual orientation. |

## 5. Discussion

From this work, it is possible to conclusively show that a Conversational AI cannot detect offensive language and/or unethical behavior, however there is a need to identify offensive language and/or unethical behavior as described below:

1.  Conversational AI models are based on foundation models trained on large, scaled data and can be adapted to a wide range of tasks. These models are not new and they have existed for decades; however, their use has increased in recent years and has brought discussions about what is possible when using these models and also understanding their characteristics;
2.  There is also a need to addresses classical ethical methodologies in algorithm design considerations for autonomous and intelligent systems (A/IS) where ML may or may not reflect ethical results considered in human decision making;
3.  It is also important for a Conversational AI to address cultural aspects involving preferences of individuals, among them global and individual preferences and cultural clusters in the use of systems;
4.  Those systems have also potential to harm historically underrepresented or disadvantaged groups of a population, because it is based on historical data often intrinsically discriminatory;
5.  Systems need to be improved to address individual justice, between the improvements, for example, identify how a pair of individuals could be treated similarly. In addition, even with bias, it is necessary systems that could provide accurate predictions for future decision-making, but without discriminating people into population subgroups;
6.  The algorithms need also to deal with changes in joint data over time, integrating an algorithm-level solution for a fair classification and an online approach to keep an accurate and up-to-date classifier to infinity data streams with non-stationary distribution and bias discrimination. It is not possible for a Conversational AI to rely on annotated data as it would break the flow of the algorithm;
7.  Domain experts are also required because NLP have complex tasks, such as questioning and answering and object recognition, using sentences or images as inputs, and it is necessary to write domain specific-logic to convert raw data into higher-level features;

8. Foundation models may be also have a cultural-centric metric by default, which may not be beneficial in other contexts where the foundation model might be applied. Furthermore, the application of these foundation models in different domains can be a force for epistemic and cultural homogenization;

9. Recent studies about NLP have been focused on gender and race bias, however there are few studies on religious bias. Therefore, it is necessary that some probes detect that those systems are not introducing any unethical behavior and/or offensive language in the areas of prompt completion, analogical reasoning, and story generation involving religious bias; and

10. NLP needs to support diverse documentation, to allow data curation, identification of potential biases and, transparency on limitations of data sets. Furthermore, it needs to support mechanisms for safe maintenance and data sharing to correct undesirable behavior due to changes in the data.

In the area of offensive language, this article presents some studies applied to text analysis, as described below.

1. The study of feelings and emotions using text analysis, as demonstrated in the study *"Linguistic Inquiry and Word Count (LIWC)"* [40];

2. In psychometric scale to measure feelings as in the study PANAS-t (*"Panas-t: A pychometric scale for measuring sentiments on twitter"* [41] and POMS-ex (*"Modeling public mood and emotion: Twitter sentiment and socioeconomic phenomena"* [42]);

3. In the study of words and expressions, as in the study conducted by the Federal University of Minas Gerais, Brazil, in the article *"A Measurement Study of Hate Speech in Social Media"* [29];

4. In GPT-3, as it can generate hateful text, its capacity could also be used to identify and classify hate speech, such as in the study from Chiu et al. [30], *"Detecting Hate Speech with GPT-3"*;

5. This work is an extension of a previous publication, "A Hybrid Dictionary Model for Ethical Analysis" [4], that proposes a dictionary template for sorting comments that may contain offensive language using a word filter. The dictionary template is not an NLP algorithm and there are some limitations when applying this dictionary to the task of identifying offensive language because it operates at the level of words, without any context. Breaking sentences into words, valuable semantic context, and content is lost. Additionally, offensive language can be used ironically, a common practice on the internet, where jokes, slang, and sarcasm are pervasive.

In the Results section, it was demonstrated the application of the Dictionary to a text corpus, in this case applied to Twitter posts. Some of the outcomes are as follows:

1. The article seeks to show that Conversational AI cannot detect unethical behavior and/or offensive language and it is also expensive and difficult to manually identify offensive posts. Furthermore, by demonstrating the application of the dictionary to a text, in this case Twitter posts, a low-level proof-of-concept is presented to detect offensive language. However, it will be necessary to refine the study to something more relevant to Conversational AI seeking more solid results such as training an ML in examples of offensive language and/or unethical behavior and then applying it as a detector;

2. For the low-level proof of concept, some sites that could contain posts with offensive language were annotated manually. Moreover, the dictionary categories were selected by checking areas that may be related to online crime, from state bodies, such as the Police, or from associations that receive reports of crime on the Internet. Furthermore, to define categories is necessary to understand cultural aspects and the law in the country or state were the template dictionary will be designed, because while some categories are clearly unethical like "rape", others are not, such as "legal drugs", because an abusive consumption could lead to offensive language, and, in Brazil, where the template dictionary was applied, some types of "gambling" are illegal;

3.  In addition, using a labeling process for offensive language and/or unethical behavior is not acceptable to the industry as they are looking to eliminate this intervention. Furthermore, a manual process would break the flow of conversational AI; and

4.  Although there is a partial applicability of the dictionary model, the results propitiated identify the complexities of categorizing offensive language in Portuguese, and demonstrates the contextual complexity including the importance of addressing bias and discrimination as well. Without proper categories, for example, the automated system could erroneously consider many people to be hate speakers and the automated system could fail to differentiate between commonplace offensive language and serious hate speech. Furthermore, other categories of offensive language could be erroneously addressed.

## 6. Conclusions

This paper focused on investigating if Conversational AI (NLP, Chatbots) could identify offensive language and/or unethical behaviors and the readiness of current foundation algorithms. This study followed a systematic mapping method to present an overview of a research area, and to report the amount and type of literature and results that are published in it.

Three mapping questions were created to investigate why a conversational AI can not detect offensive language and/or unethical behavior. Overall, six classification criteria were used to analyze the articles, including type of research, empirical type, type of techniques, source of publication, and conferences.

The research resulted in 24 articles selected from the fields of AI Ethics, Machine Learning (ML) Fairness, Language Models, and solutions to detect offensive language and/or unethical behavior in the Internet.

From this literature, we conclude that Foundations Models used in Conversational AI requires better understanding of their characteristics, as also probe and more evidence that these models can deal properly with gender, racial, religious bias, and other biases and their potential to harm historically under-represented or disadvantaged groups of a population.

The study also applies a low-level proof of concept of a template dictionary to filter potential offensive language in Twitter and demonstrates that design of categories for offensive language also requires understanding of cultural characteristics and law in the region where the analysis will be conducted.

### 6.1. Specific Conclusions

The study shows that, due to the heavy use of social media, it is expensive and difficult to manually identify offensive posts. Furthermore, using a labeling process to detected offensive language and/or unethical behavior is not acceptable to the industry as they are looking to eliminate this intervention. Furthermore, a manual process would break the flow of conversational AI. There are also minimum standards required to address fairness in ML, as conversational AI is built based on historical data. However, historical data could be intrinsically discriminatory. Furthermore, Conversational AI uses Foundation Language Models, and homogenization provides the use of these models in various tasks and therefore introduces justice and ethics problems inherited by all models and in a variety of tasks.

### 6.2. General Conclusions

Technology is becoming a commodity and it is important that ethical design starts at the beginning of a scientific research of AI algorithms.

Language models are difficult to understand and may have unexpected flaws. Mitigating risks has become one of the central tasks in the development of foundation models from the ethical and safety perspective of AI. Moreover, homogenization of foundation

models has the potential to amplify bias and injustices rather than distributing them, in addition to increasing exclusion.

Furthermore, the limits of acceptability are rapidly changing and the concept of justice in data-driven societies requires further studies by researchers. This work presents practical guidance on how Conversational AI algorithms work with the ethical issues and minimum principles for ethics and fairness in ML algorithms.

### 6.3. Recommendations

This article presents an overview about the fields involving AI Ethics, ML Fairness, Offensive Language, and Conversational AI, and recommends the following further studies to expand the understanding of ethical questions related to Foundation Language Models:

(1) New investigations to show more areas impacted by biases, analyzing how large language models could produce quite a bit of unethical behavior because of bias in the historical data; (2) new investigations on methods for a fair ML applied specifically to Conversational AI; (3) new studies on how conversational AI deal with cultural clusters; (4) new investigations in the context of irony in offensive language because in the Internet, jokes, slang, and sarcasm are pervasive; (5) new studies analyzing if introduction of positive associations could mitigate bias in Conversational AI; and (6) new studies analyzing data sheets and/or data statements for NLP solutions and how those systems could adapt to dynamic changes in the data and mitigation of bias.

### 6.4. Future Work

As a next step in this research, we aim to refine the study to something more relevant to Conversational AI than a template dictionary, seeking more solid results such as training an ML in examples of offensive language and/or unethical behavior and then applying it as a detector.

In particular, discussions about language model ethics needs to be stimulated in Latin America to respond to specific language needs and social characteristics. The region has a large pool of people with soft-skills, problem solving, young people, collaborative, and creative leadership to drive the ethical use of AI. Furthermore, there are a big cultural cluster and regional preferences need to be investigated as well identified in which areas we converge due to geographic proximity.

**Author Contributions:** For this research there are equally contributing authors who designed the concept of the research; implemented the experimental design; conducted data analysis; prepared the draft paper; reviewed the whole paper. All authors have read and agreed to the published version of the manuscript.

**Funding:** General and financial support during this investigation: the Ecossistema Negócios Digitais Ltda; the Brazilian Aeronautics Institute of Technology (Instituto Tecnológico de Aeronáutica—ITA); the Casimiro Montenegro Foundation (Fundação Casimiro Montenegro Filho—FCMF), and The Higher Level Personnel Improvement Coordination (Coordenação de Aperfeiçoamento de Pessoal de Nível Superior—CAPES).

**Institutional Review Board Statement:** Not applicable.

**Informed Consent Statement:** Not applicable.

**Data Availability Statement:** Data sharing not applicable.

**Acknowledgments:** The authors gratefully thank to: the Ecossistema Negócios Digitais Ltda; the Brazilian Aeronautics Institute of Technology (Instituto Tecnológico de Aeronáutica—ITA); the Casimiro Montenegro Foundation (Fundação Casimiro Montenegro Filho—FCMF), and The Higher Level Personnel Improvement Coordination (Coordenação de Aperfeiçoamento de Pessoal de Nível Superior—CAPES), for all their general and financial support, during this investigation.

**Conflicts of Interest:** The authors declare no conflict of interest.

## Abbreviations

The following abbreviations are used in this manuscript:

| | |
|---|---|
| ACM FAT | ACM Conference on Fairness, Accountability, and Transparency |
| AGI | Artificial General Intelligence |
| AI | Artificial Intelligence |
| AAAI | Association for the Advancement of Artificial Intelligence |
| A/IS | Autonomous and Intelligent System |
| ASR | Automatic Speech Recognition |
| BERT | Bidirectional Encoder Representations for Transformers |
| EAD | IEEE Ethically Aligned Design |
| GPT | Generative Pre-trained Transformer |
| HAI | Stanford Human-Centered Artificial Intelligence Laboratory |
| NLP | Natural Language Processing |
| ICML | International Conference on Machine Learning |
| IEEE | Institute of Electrical and Electronics Engineers |
| NeurIPS | Neural Information Processing Systems |
| NLU | Natural Language Understanding |
| SLU | Spoken Language Understanding |

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
