# Peer review of "Could a Conversational AI Identify Offensive Language?â€"

_information, doi:10.3390/info12100418_

Round 1
Reviewer 1 Report
Do the ethical issues depend on the geographical area? Should it be considered part of the process?Author Response
Dear Sir/Madam,
We appreciate your feedback and valuable comments. See in the PDF attached our disposition for the comments received by the reviewers, in red for the text removed and in blue for the text added. We would also like to highlight our appreciation for your comments, as they were essential to review the article and improve our line of research.
Regarding the content of the article, we made considerable changes, and kept around 50% of the original content related to our study on using Conversational AI to identify offensive language. And 50% of the changes refer to removal of introductory topics on AI and Natural Language Processing that are not relevant for the readers of this journal, and inclusion of new topics more relevant to Conversational AI such as classical Ethics in AI, Fair Machine Learning, Foundation Language Models and applications to detect Offensive Language, as
recommended by the reviewers.
Please find a attached letter with the revision of your comments and suggestions. We hope that this version is in better quality and favorable to publication.
In advance, we would like to thank you for your collaboration and appreciation of our work.
Yours Sincerely,
Daniela América da Silva - Corresponding Author
September 2021

Reviewer 2 Report
This paper focuses on the unethical behaviors of Conversational AI. It is claimed that the readiness of current algorithms for human dialogue as well as the identification of their unethical behaviors are investigated. In order to improve the quality of the paper, the authors should pay more attention to the following suggestions:
1. The title and abstract of the paper do not reflect the main focus of this study, i.e., the paper including its presented experiments mainly discusses the unethical behaviors of conversational AI rather than the identification and mitigation of bias of conversational AI. It is also suggested that the authors should tone down their relevant claims. In addition, as this work is an extension of a previous publication, a “summary of difference” file and relevant statement within the paper detailing how this work provides a significant contribution beyond the previous publication is expected.
2. Many parts of the paper read more like a storytelling or news report rather than a scientific paper. For example, most of the paragraphs of the introduction section discuss the historical evolution of AI. It is expected that state of the art fair conversational AI to be discussed at the minimum. So do sections 2.3, 2.4, 2.5 and so on, which do not help with the claimed contributions of this work. It is also a stretch to argue that related work of ethical AI in general is missing, such as
Delayed impact of fair machine learning. ICML 18
Faht: An Adaptive Fairness-aware Decision Tree Classifier. IJCAI 19
Metric-Free Individual Fairness in Online Learning. ICML 20
Farf: a fair and adaptive random forests classifier. PAKDD 21
The authors should include these recent works when discussing ethical AI then narrow down to ethical conversational AI for a comprehensive literature review.
3. A thorough proofread is expected to fix presentation issues. For example, “human conversations has...”.
Author Response
Dear Sir/Madam,
We appreciate your feedback and valuable comments. See in the PDF attached our disposition for the comments received by the reviewers, in red for the text removed and in blue for the text added. We would also like to highlight our appreciation for your comments, as they were essential to review the article and improve our line of research.
Regarding the content of the article, we made considerable changes, and kept around 50% of the original content related to our study on using Conversational AI to identify offensive language. And 50% of the changes refer to removal of introductory topics on AI and Natural Language Processing that are not relevant for the readers of this journal, and inclusion of new topics more relevant to Conversational AI such as classical Ethics in AI, Fair Machine Learning, Foundation Language Models and applications to detect Offensive Language, as recommended by the reviewers.
Please find a attached letter with the revision of your comments and suggestions. We hope that this version is in better quality and favorable to publication.
In advance, we would like to thank you for your collaboration and appreciation of our work.
Yours Sincerely,
Daniela América da Silva - Corresponding Author
September 2021

Reviewer 3 Report
The premise of the paper -- Can conversational AI (NLP, chatbots) notice unethical behavior? -- is a good question, and could lead to an interesting paper. But the structure of the paper is poor, and the investigation carried out is mostly irrelevant to the main question and title of the paper.
Most broadly, the paper is not about unethical "behavior" - it is about offensive language use -- user behavior more generally is not discussed. The choice of calling such language "unethical" raises questions about who would describe the language as unethical. The authors give no rationale for why they chose the categories of unethical language they did. While categories like "rape" are clearly unethical, others, such as "legal drugs" and "gambling," reflect the authors' values more than an objective assessment of ethicality.
After finding that state-of-the-art AI systems cannot identify "unethical behavior" (again, I would say, "offensive language"), the authors propose a template for a dictionary of unethical behavior. This is a rather outdated solution that could hardly be considered NLP. Moreover, the authors claim the dictionary includes "sentiment analysis" but do not in fact describe that aspect of the project in this paper. Ultimately, the dictionary system described is inadequate to the task of identifying "unethical behavior" for one simple reason: it operates at the level of words, without any context. Indeed, by breaking sentences into words, valuable semantic context and content is lost. Because "offensive language" can be used ironically--a common practice on the internet, where jokes, slang, and sarcasm are pervasive--a tool that identifies potentially offensive words without any context is nearly useless.
As the authors discuss, companies like FaceBook already have AI tools for identifying a high percentage of offensive content ("...in Q4 2020 the AI had a proactive detection of 97.1% of hate speech posts"). This seems like a promising approach that could be extended to conversational AI, which appears to be the authors' intention in this paper. But if so, their dictionary method is has a long way to go before it could be considered an NLP tool or "conversational AI" for identifying "unethical behavior" or "offensive speech." To rely on a human to double-check words flagged by the dictionary would completely break down the conversational flow and render the chatbot useless.
In addition, the authors don't spend much time grappling with the meaning of "unethical" -- this is a fluid concept that changes over time and according to the society in which the tool is used. They seem to unreflectively adopt their own idiosyncratic definition of "unethical" and apply that to the dictionary.
More specifically, the introduction section is mostly irrelevant and should be condensed to 3 paragraphs (at most) that 1) introduce AI as a technology widely used on the internet and elsewhere 2) describe NLP and ML as important subfields that have made Chatbots and conversational AI possible, but 3) raise questions about whether and how they can cope with unethical user behavior. All discussion of Babbage and Turing can be profitably removed, as it will not be novel to readers of this journal.
The section "Methods" is mislabeled. It should be called "Background" and condensed from 13 pages to 3 at most. Instead of offering a sprawling literature review of AI ethics, NLP, transformers, etc., the section should slice through these literatures to show conclusively that "Conversational AI cannot detect unethical behavior (offensive language)." The authors might consider including more recent work that shows how large language models often produce quite a bit of "unethical behavior" (e.g.)
https://www.nature.com/articles/s42256-021-00359-2
http://arxiv.org/abs/2108.07258
The Background section should demonstrate the need for "unethical behavior detection" in conversational AI (currently doesn't do this clearly). Then there is a rationale for the project that follows.
Then the paper should turn to the "Methods" section where the authors describe the creation of the dictionary -- not the method of the literature review.
In the "Results" section, the authors should describe the application of the Dictionary to a text corpus. In the current paper, the authors apply the dictionary to Twitter posts. This may work as a low-level proof of concept, but a target more relevant to Conversational AI would make for stronger results.
The "discussion" section should focus on the results of this paper in light of the research landscape as described in the "background" section. At present the discussion is too broad.
Finally, the "conclusion" section should only make conclusions that can be drawn from the work conducted in this paper. As it is, the authors make a great many recommendations that have little to do with the work they conducted.

Author Response

(The authors gave the same response as above.)

Round 2
Reviewer 2 Report
My previous revision comments have not been fully addressed:
1) The main challenges of the existing approaches should be discussed to motivate this study then detailing how this study advances the state of the art.
2) The suggested recent ethical AI works are for example purpose and are not inclusive. It is expected that the discussion also represents the development of different approaches rather than simply stating and adding more, e.g., “method 1 addresses... then to improve the drawbacks of method 1, method 2…”. The authors can use these papers as seeds of reference instead, and keep researching respective authors' other fairness relevant works as well as references provided therein for a comprehensive literature review. Similar strategy is suggested to motivate and solid the necessity of this work.
3) To distinguish this work from the previous publication from experimental results’ perspective or discuss the inapplicability of the previous method if not comparable.
Author Response
Dear Sir/Madam,
We appreciate your feedback and valuable comments. See our disposition in red, for each comment. We would also like to highlight our appreciation for your comments, as they were essential to review the article and improve our line of research.
Regarding the content of the article, we made minor changes, firstly improving Introduction and Background, adding new references including a new subsection about Bias and Discrimination. Secondly, in the section Methods is clarified partial contribution from previous publication and added in the section Discussion the explanation of how this article could propitiate advances in the field.
We hope that this version is in better quality and favorable to publication.
In advance, we would like to thank you for your collaboration and appreciation of our work.
Yours Sincerely,
Daniela América da Silva - Corresponding Author
October 2021

Reviewer 3 Report
Significantly re-written and moderately improved since the first draft. I note in several places that suggestions I made in the first review were addressed, so thank you for responding. I think it is overall a better paper, but I have some concerns:
- The paper does not make clear how it improves upon earlier work. As you say, GPT-3 is able to identify unethical behavior with accuracy up to 78%. So in answer to the title of the question of the paper, the answer would seem to be, Yes, with 78% accuracy. If this is the case, what is the significance of the dictionary you create? Or are you not considering GPT-3 to be a "conversational AI"? Clarification on these points would help.
- Similarly, you say Facebook has a 98% accuracy in identifying hate speech. I understand that Facebook is not conversational AI, but your paper doesn't make clear how your dictionary solution would be used by a Conversational AI to identify unethical speech. Some clarification on this point would be helpful.
- Step 3 of Literature Review in Background section doesn't explain the criteria used to select articles and books from the many that were retrieved by search. More details as to why those articles were chosen would be helpful. Also, the works cited in background section of this version of the manuscript have changed considerably since last time, making me wonder if the description of the selection process is still relevant.
- Section 2.1 = abbreviation still misspelled
- In multiple places it is stated that "it is practically impossible to have an army for post and video analysis" -- but this is in fact what Facebook does. Recent estimates suggest they employ over 30,000 people to do just that. So a reframing of this point would make more sense: e.g. it is expensive and difficult to manually identify offensive posts, etc.
- The lists of words on pages 9-10 are unnecessary and can be reduced to 3 examples each for "Hate," "Intensity," etc.
- If this study is an extension of the paper, “A Hybrid Dictionary Model for Ethical Analysis,” is it really necessary to include the description of creating the dictionary model? Can't the earlier paper be referenced? This paper is nominally about Conversational AI -- but there is no discussion of how a dictionary model relates to conversational AI. At the very minimum, the paper should explain how such a dictionary system could be implemented in theory, if not in practice.
Author Response
Dear Sir/Madam,
We appreciate your feedback and valuable comments. See our disposition in red, for each comment. We would also like to highlight our appreciation for your comments, as they were essential to review the article and improve our line of research.
Regarding the content of the article, we made minor changes, firstly improving Introduction and Background, adding new references including a new subsection about Bias and Discrimination. Secondly, in the section Methods is clarified partial contribution from previous publication, added that recent advances in offensive langugage identification are not clearly explaining how it is dealing with bias and discrimination aspects, and added in the section Discussion the explanation of how this article could propitiate advances in the field.
We hope that this version is in better quality and favorable to publication.
In advance, we would like to thank you for your collaboration and appreciation of our work.
Yours Sincerely,
Daniela América da Silva - Corresponding Author
October 2021
